# Taste bud formation depends on taste nerves

**Di Fan[1,2], Zoubida Chettouh[1], G Giacomo Consalez[3], Jean-François Brunet[1]\***

[1]Institut de Biologie de l'ENS (IBENS), Inserm, CNRS, École normale supérieure, PSL Research University, Paris, France; [2]School of Life Science, East China Normal University, Shanghai, China; [3]San Raffaele Scientific Institute and Università Vita-Salute San Raffaele, Milano, Italy

**Abstract** It has been known for more than a century that, in adult vertebrates, the maintenance of taste buds depends on their afferent nerves. However, the initial formation of taste buds is proposed to be nerve-independent in amphibians, and evidence to the contrary in mammals has been endlessly debated, mostly due to indirect and incomplete means to impede innervation during the protracted perinatal period of taste bud differentiation. Here, by genetically ablating, in mice, all somatic (i.e. touch) or visceral (i.e. taste) neurons for the oral cavity, we show that the latter but not the former are absolutely required for the proper formation of their target organs, the taste buds.
DOI: https://doi.org/10.7554/eLife.49226.001

## Introduction

Taste buds are onion-shaped clusters of 60–100 taste receptors and support cells, embedded in epidermal papillae and distributed in a punctate pattern in the tongue and soft palate epithelia. They sense nutrients in the oral cavity and transmit taste information to the termini of sensory neurons, through conventional (*Finger et al., 2005*) and non-conventional (*Ma et al., 2018*; *Romanov et al., 2018*) synapses. Taste receptors and their support cells have a limited life span of 8 to 20 days depending on cell types (*Perea-Martinez et al., 2013*) and are constantly renewed from progenitors situated outside (*Okubo et al., 2009*; *Ohmoto et al., 2017*; *Perea-Martinez et al., 2013*) and, for a small minority, inside (*Perea-Martinez et al., 2013*) (and references therein) the taste bud. This process depends on the sensory afferents as demonstrated by the degeneration and regeneration of taste buds triggered, respectively, by the degeneration and regeneration of their surgically severed nerves (*Oakley and Witt, 2004*). While nerve-dependency has been uncontroversial for more than a century concerning the maintenance of taste buds in adult animals (*Jacobson, 1991*), it is still debated concerning their initial development in the embryo. Mature taste buds have been observed to form in the absence of nerves in amphibians (*Barlow et al., 1996*). In mammals, the case for nerve-dependency of taste bud development was mostly made through genetic invalidations of neurotrophins or their receptors, which hamper, to various extents, the outgrowth of sensory afferents towards the taste buds and simultaneously entail large losses of taste buds on the tongue at postnatal stages (*Oakley and Witt, 2004*). However, the protracted development of lingual taste buds, most of which mature after birth, left open the possibility that their postnatal loss reflected a failure of 'maintenance' or 'maturation' after a process of nerve-independent embryonic development (*Barlow and Northcutt, 1998*; *Kapsimali and Barlow, 2013*). To resolve this decades-old controversy, we completely blocked sensory innervation of the oral cavity in mouse by genetically preventing the birth of cranial sensory neurons, and examined the formation of taste buds — including palatal ones, whose development is almost entirely embryonic.

**\*For correspondence:**
jfbrunet@biologie.ens.fr

**Competing interests:** The authors declare that no competing interests exist.

# Results and discussion

Taste organs (taste papillae and their resident taste buds) of the anterior tongue and soft palate are innervated by somatic sensory neurons (for touch and pain) located in the trigeminal ganglion and visceral sensory neurons (for taste) located in the geniculate ganglion (*Watson et al., 2012*). The circumvallate papilla receives innervation from visceral sensory neurons in the distal (petrosal) ganglion of the glossopharyngeal nerve, while its somatosensory ones, projecting in the same nerve (*Frank, 1991*) (and references therein) are probably located in the proximal ('superior') ganglion. To suppress innervation of taste organs, we exploited the developmental dependency of cranial sensory neurons on the proneural transcription factors *Neurog1* and *Neurog2*. As early as day 9.5 of embryonic development (E9.5), the knockout of *Neurog1* blocks neuronal differentiation in the trigeminal, superior and jugular ganglia (*Ma et al., 1998*) which harbor somatic sensory neurons. Inactivation of its paralogue *Neurog2*, expressed in epibranchial placodes, blocks the formation of the geniculate and petrosal ganglia, which harbor visceral sensory neurons (*Fode et al., 1998*). In each single *Neurog1* and *Neurog2* knockout at E16.5, the epithelium of the tongue and of the soft palate retained nerve fascicles at regularly spaced locations (corresponding to the taste organs) from, presumably, visceral or somatic sensory fibers, respectively (which thus navigate to their target independently of each other) (*Figure 1—figure supplement 1*). In double *Neurog1/Neurog2* knockouts, which lose all sensory innervation of the head (*Espinosa-Medina et al., 2014*), all nerve fibers had disappeared from the epithelium and underlying lamina propria of the palate and tongue, as expected (*Figure 1— figure supplement 1*).

   Taste organs differentiate from taste placodes, specialized patches of the oral epithelium, thickened by apico-basal elongation of the cells, which transform into dome-shaped papillae with a mesenchymal core — most dramatically at the site of the single circumvallate papilla, on a smaller scale for fungiform papillae of the tongue, and also in the palate albeit less conspicuously (*Rashwan et al., 2016*) — and express a number of markers. Expression of the signaling molecule *sonic hedgehog* (*Shh*), after a diffuse phase throughout the oral epithelium, resolves around E12.5 into a punctate pattern corresponding to taste placodes, which also start expressing *Prox1* and high levels of *Sox2* (*Thirumangalathu et al., 2009*; *Nakayama et al., 2008*; *Liu et al., 2013*; *Okubo et al., 2006*). Two days later, *Shh*, *Sox2* and *Prox1* were still expressed in the same pattern in the placodes — by then about to become papillae (hereafter placodes/papillae) — (*Figure 1A,B*, *Figure 1—figure supplement 2*), and the transcription factors *Ascl1* and *Hes6* (*Seta et al., 2003*) were switched on in just a few cells (*Figure 1B*, *Figure 1—figure supplement 2*). All these markers were expressed in a normal pattern in the soft palate and anterior tongue of double *Neurog1/2* knockouts at E14.5, except for *Sox2* whose expression was stronger and expanded in the palate (*Figure 1C*, *Figure 1—figure supplement 2*). Expression of these markers was not restricted to taste placodes/papillae but also occurred in the incipient ridges (rugae) of the hard palate (which never give rise to taste buds) and this expression was also essentially unchanged in *Neurog1/2* knockouts (*Figure 1A,D*). Between E14.5 and E16.5, a cluster of cells in each taste papilla or ruga had switched on *cytokeratine 8* (*CK8*) in both wild type and *Neurog1/2* KO (*Figure 1B,D*, *Figure 1— figure supplement 2*). Thus, fungiform and palatal taste papillae (whose morphology and, as we show here, gene expression program are similar to that of palatal rugae) are epithelial specializations that form in the absence of any nerve, in agreement with prior observations of fungiform placodes/papillae development in cultured tongue explants (*Farbman and Mbiene, 1991*, *Mbiene et al., 1997*, *Hall et al., 2003*). The single circumvallate papilla of the posterior tongue stood in contrast. In the wild type, it displayed the same gene expression events as fungiform and palatal papillae on its dorsal surface (*Figure 1A,E*). However, in this case, as in rugae, expression of *Shh*, Sox2, *Hes6*, *Ascl1*, *Prox1* and *CK8* does not prefigure the later differentiation of taste bud cells, which takes place after birth, mostly in the semi-circular trenches, not at the dorsal surface. In the *Neurog1/2* knockouts at E14.5 (a day after arrival of nerve, thickening of the placode and onset of *Shh* expression (as previously observed on tongue explants [*Mistretta et al., 2003*] and see *Figure 1—figure supplement 3A*), the sparse expression of *Hes6*, *Ascl1* and *Prox1* was preserved, but *Shh* and *Sox2* was not upregulated and morphogenetic events leading to papilla formation were stalled (*Figure 1F*), corroborating and extending a previous observation of circumvallate papilla atrophy in *Brain-Derived Neurotrophic Factor/Neurotrophin three* double knockouts (*Ito et al., 2010*). A similar phenotype was obtained in single *Neurog2* KO (*Figure 1—figure supplement 3A,B*), which lack

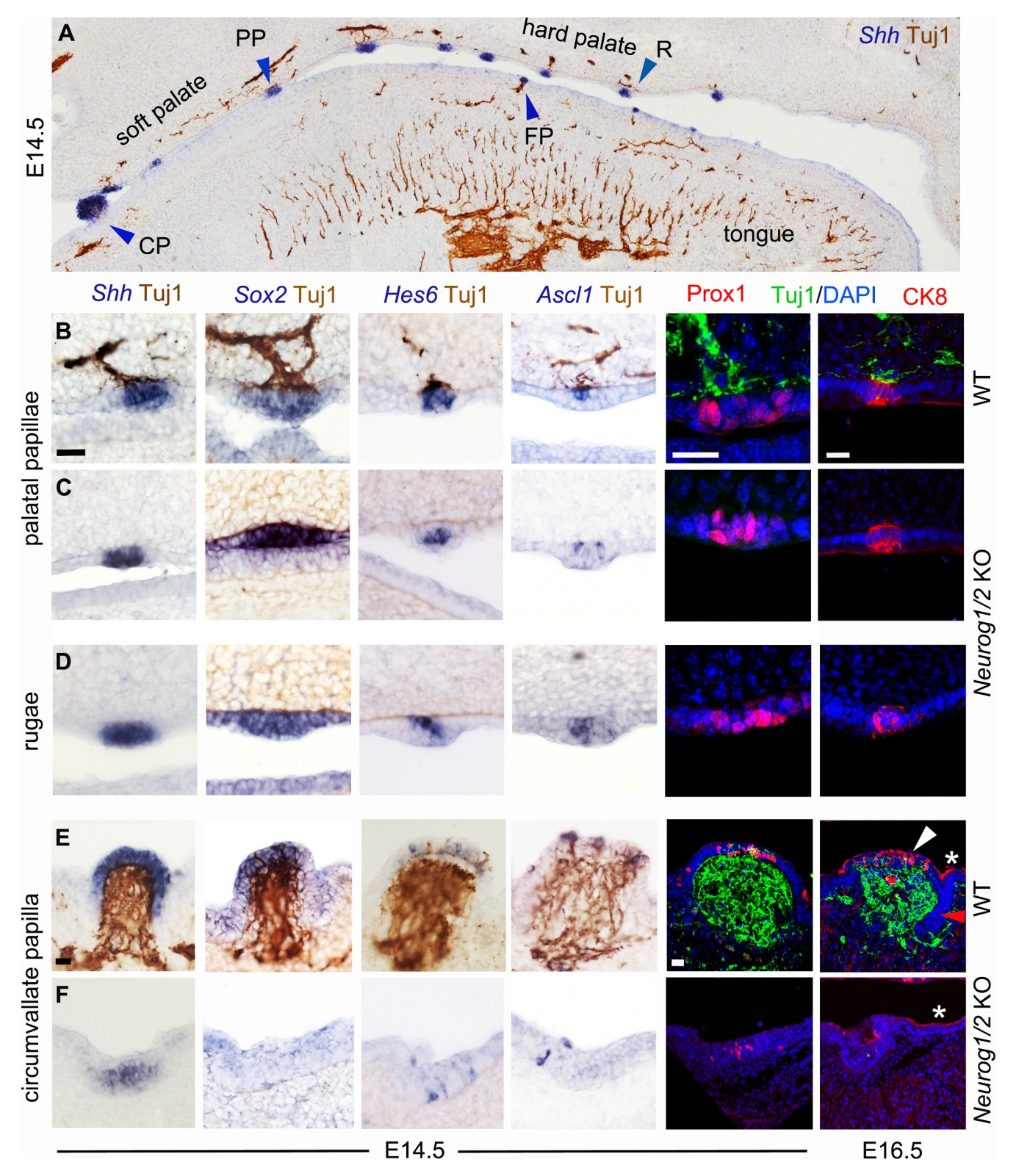

**Figure 1.** Soft palate taste papillae as well as hard palate rugae, but not the tongue circumvallate papilla form without innervation. (A–F) Combined immunohistochemistry for β-III tubulin (Tuj1, brown) and in situ hybridization (blue) for the indicated probes (top panel and left four columns), or immunofluorescence for Prox1 or CK8, combined with immunofluorescence for β-III tubulin and a counterstain with DAPI (two right columns) in wild type (A,B,E) and double *Neurog1/2* KO (C,D,F) at E14.5 or E16.5 as indicated. In the circumvallate papilla, markers are expressed on the dorsal surface

*Figure 1 continued on next page*

*Figure 1 continued*

(white arrowhead for CK8) but not at the lower part of the trenches (red arrowhead in the right column), where taste buds will develop after birth. CK8 is also expressed in flattened cells of the periderm (asterisks). For every probe two animals were examined. CP : circumvallate papilla; FP : fungiform papilla ; PP : palatal papilla ; R : ruga. Scale bars: 20 μm.

DOI: https://doi.org/10.7554/eLife.49226.002

The following figure supplements are available for figure 1:

**Figure supplement 1.** Sensory innervation of the oral cavity in *Neurog1*, *Neurog2* and double *Neurog1/Neurog2* knockouts.

DOI: https://doi.org/10.7554/eLife.49226.003

**Figure supplement 2.** Placodal/papillary gene expression is nerve-independent in fungiform papillae.

DOI: https://doi.org/10.7554/eLife.49226.004

**Figure supplement 3.** The circumvallate papilla does not form in the absence of visceral innervation.

DOI: https://doi.org/10.7554/eLife.49226.005

petrosal ganglia (*Figure 1—figure supplement 3C*). Therefore, the circumvallate papilla, already known to differ from fungiform papillae by its ontogenetic requirement for an *Fgf10* signal from the mesenchyme (*Petersen et al., 2011*), also differs from both fungiform and palatal papillae by requiring its afferent nerve for its formation.

By E18.5, in wild type embryos, taste buds had begun to form, as onion shaped clusters of many cells expressing *CK8*, spanning the height of the oral epithelium, to different extents in different locations. From this developmental stage on, we will designate these taste bud anlagen, irrespective of their size and degree of maturation, as 'CK8-positive (CK8$^+$) cell clusters'. In the palate, they almost reach their mature size and structure by E18.5 (*Rashwan et al., 2016*) (*Figure 2A*). In *Neurog2* knockouts, that is in the absence of visceral nerves, there was a dramatic deficit in both the number of CK8$^+$ cell clusters detected in every other section (43% fewer) in the palate (this number can be taken as an approximation of the number of taste buds in the palate, see Materials and methods) and their cross-sectional area (66% smaller) so that the overall area occupied by CK8$^+$ cells was 80% smaller than in wild types (*Figure 2A*). The latter measure, on two-dimensional projections of confocal stacks through 20 μm-thick sections, likely underestimates the extent of the deficit. To rule out that the massive shortfall in taste cells in *Neurog2* KO at E18.5 reflected developmental delay rather than failure, and to circumvent the neonatal death of *Neurog2* KO, we prolonged pregnancies by 2 days with Delvosteron delivered to the dams, and examined the embryos at E20.5 (equivalent to post-natal day 2 (P2) for the normal pregnancy of C57BL/6 mice). The degree of atrophy was even more pronounced at E20.5 than at E18.5, due to a decrease in the number of CK8+ cell clusters (now 89% fewer than in wild type) and a stagnation of their average size (while their wild type counterparts had enlarged), so that the total area occupied by CK8$^+$ cells in the soft palate was smaller by 96% relative to wild type (*Figure 2B*). CK8$^+$ cell clusters in the tongue, which are harbored in fungiform papillae and are still immature at this stage in wild type, were similarly atrophic in the mutants (*Figure 2—figure supplement 1*). Those of the circumvallate papilla were absent (*Figure 2—figure supplement 1*). To verify that the taste bud phenotype of *Neurog2* knockouts does not stem from a cell-autonomous action of *Neurog2* in taste placodes (through some undetected transient expression there), we devised a second genetic strategy to prevent visceral innervation of the soft palate. We found that ablating the transcription factor *Foxg1*, expressed in epibranchial placodes (*Hatini et al., 1999*), causes defects in epibranchial ganglia, and a fully penetrant (n = 5) lack of the greater superficial petrosal nerve as early as E11.5 — thus of the visceral innervation to the soft palate (*Figure 2—figure supplement 2*). In *Foxg1* KO embryos at E18.5 the palatal taste buds were atrophic in a manner comparable to *Neurog2* KO embryos (*Figure 2—figure supplement 2*). In contrast to visceral (taste) innervation, somatic (touch) innervation was dispensable for taste bud formation, as demonstrated by the normal number of palatal CK8$^+$ cell clusters and total area occupied by CK8$^+$ cells in E20.5 *Neurog1* knockouts (*Figure 2C*).

At E20.5 (P2), many palatal taste buds in wild types are fully mature and probably functional (*Rashwan et al., 2016*). In line with this, we could detect type I, II and III cells recognizable by the expression, respectively, of the ecto-ATPase *Entpd2*, the Gustducin α-chain *Gnat3* or the cation channels *Trpm5* and *Pkd2l1*, all cells sharing the expression of the potassium channel *Kcnq1* (*Matsumoto et al., 2011*) (*Figure 3A*). In *Neurog2* knockouts, expression of each marker was missing in many of the residual CK8$^+$ clusters (*Figure 3B*), but present, except for *Pkd2l1*, in one or two

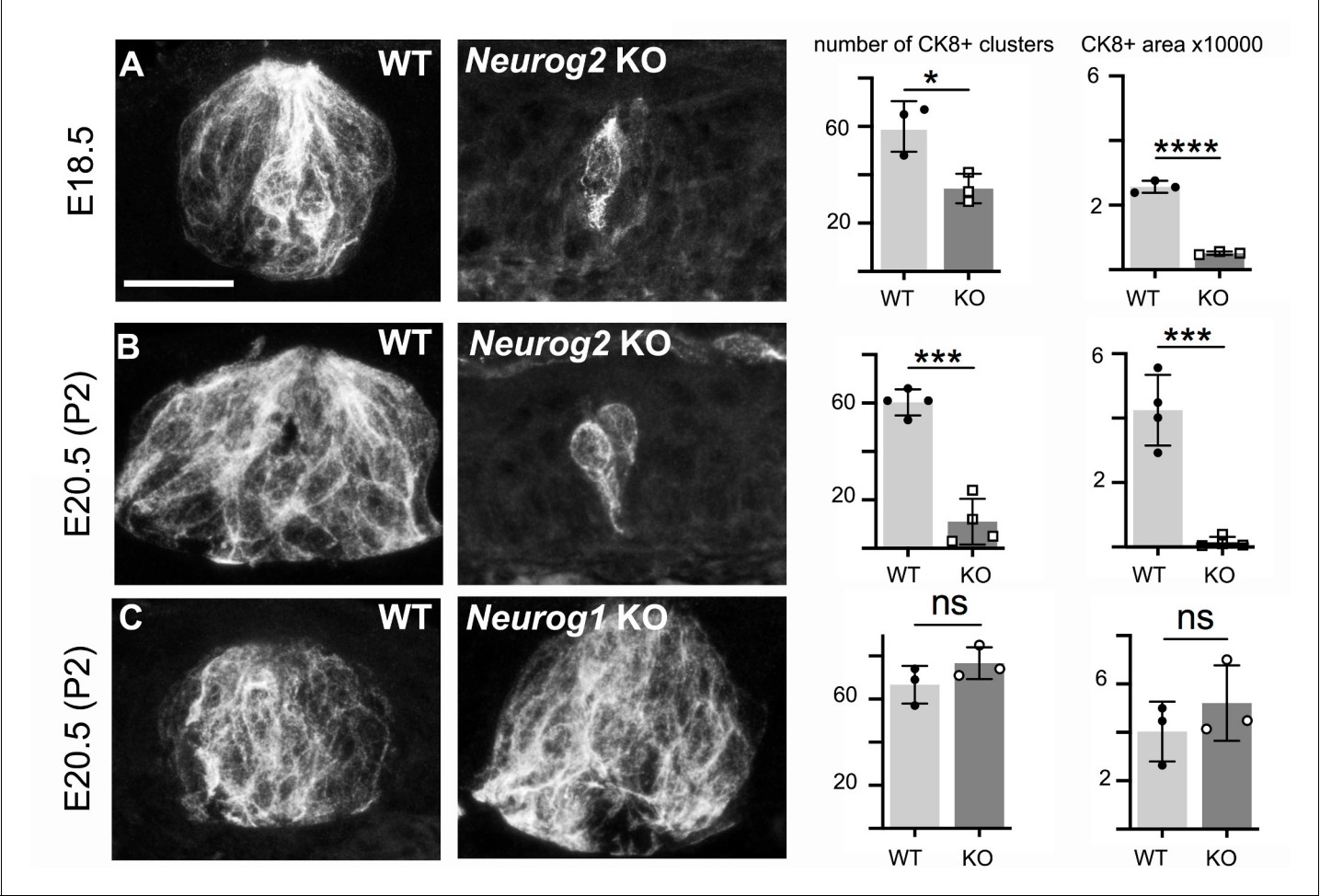

**Figure 2.** Formation of palatal taste buds requires visceral but not somatic innervation. Typical examples of CK8+ cell clusters detected by immunofluorescence in wild type and *Neurog2* or *Neurog1* KO, at E18.5 (**A**) or E20.5 (**B,C**), and quantification of the number of clusters throughout the soft palate on alternate sections, and overall surface (in $\mu m^2$) occupied by CK8$^+$ cells (in alternate sections) throughout the soft palate. *: p<0.05, ***: p<0.001, ****: p<0.0001 and ns: p>0.05; error bars presented as mean ± SD (n = 3 for **A, C** and n = 4 for **B**). The individual results for each animal are represented by dots. Scale bar: 20 $\mu m$.

DOI: https://doi.org/10.7554/eLife.49226.006

The following source data and figure supplements are available for figure 2:

**Source data 1.** Cross-sectional area (in $\mu m^2$) occupied per taste bud.

DOI: https://doi.org/10.7554/eLife.49226.009

**Figure supplement 1.** CK8$^+$ cell clusters are atrophic in fungiform papillae and absent in the circumvallate papilla of *Neurog2* KO.

DOI: https://doi.org/10.7554/eLife.49226.007

**Figure supplement 2.** Taste buds do not form in the soft palate of *Foxg1* knockouts.

DOI: https://doi.org/10.7554/eLife.49226.008

cells of others (*Figure 3C*). *Neurog1* KO pups had a normal complement of differentiated cells in their fully formed buds (*Figure 3D*). The cells in the *Neurog2* mutants that display one marker or other of mature taste receptors argue that taste nerves are indispensable, not for the differentiation of CK8$^+$ cells into taste bud cells (by the criterion of the markers tested), but for the constitution, maintenance or proliferation of the pool of progenitors required for bud formation, possibly reminiscent of the role of parasympathetic nerves in the organogenesis of salivary glands (*Knox et al., 2010*). In line with this, high *Sox2* expression in perigemmal cells (i.e. closely apposed to the taste buds), a hallmark of taste cell progenitors (*Okubo et al., 2006*; *Ohmoto et al., 2017*), was massively reduced in *Neurog2* KO (although a few residual CK8$^+$ cells were themselves Sox2$^+$) (*Figure 4A*), together with that of the proliferation marker Ki67 (*Figure 4B*). We could not detect any sign of cell

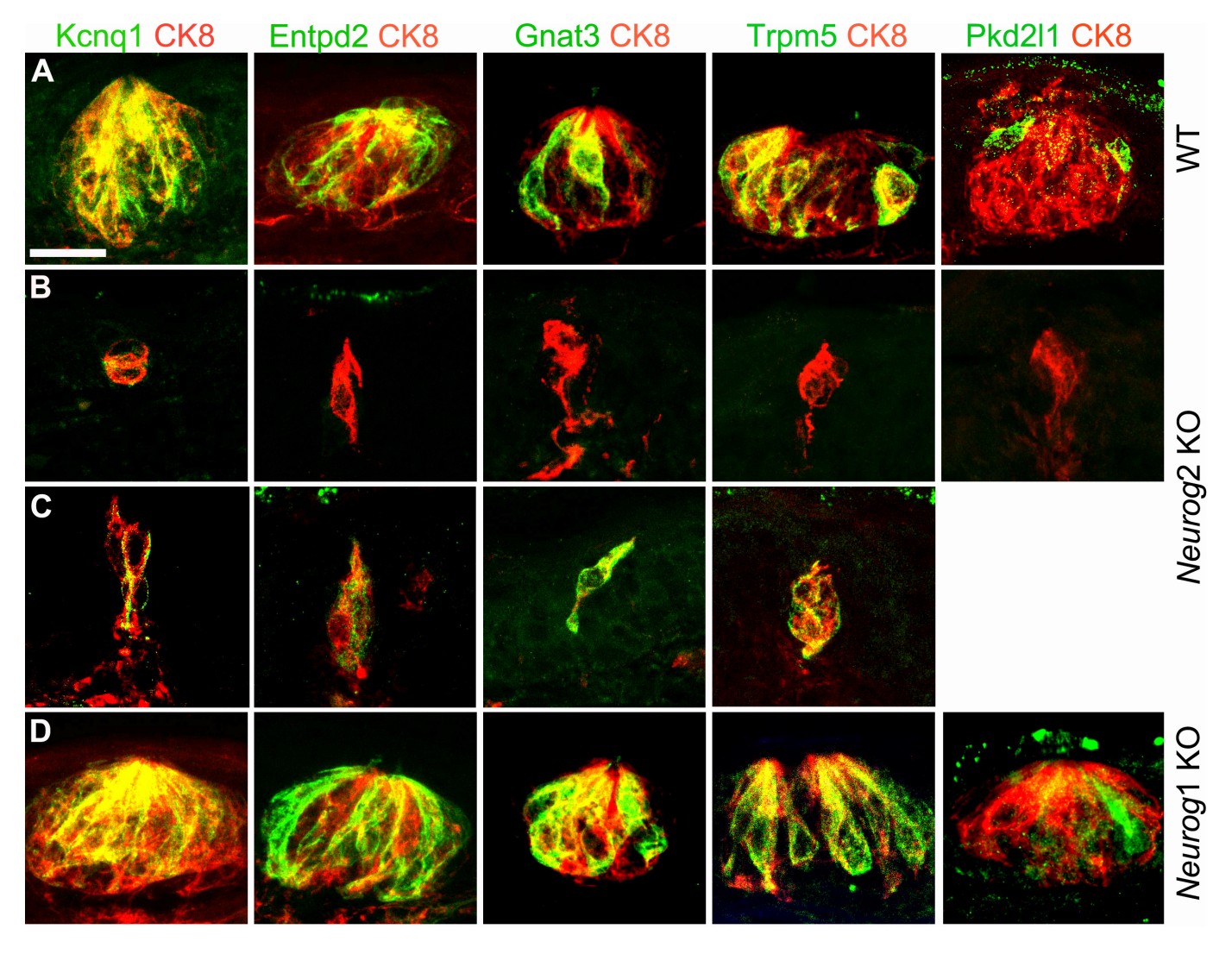

**Figure 3.** Taste bud cell differentiation is impeded, but not always, in the absence of visceral innervation. Sections through taste buds in the soft palate of wild types (**A**), *Neurog2* KO (**B,C**) and *Neurog1* KO (**D**) at E20.5 (P2), immunostained with the indicated antibodies. *Kcnq1*, *Entpd2*, *Gnat3* and *Trpm5* are occasionally detected in residual CK8+ cells of *Neurog2* KO embryos. In *Neurog1* KOs, taste buds contained the normal complement of markers. Scale bar: 20 μm.

DOI: https://doi.org/10.7554/eLife.49226.010

death at E16.5 or E18.5 in or around the taste cell clusters of wild type or mutant pups by the TUNEL reaction or immunofluorescence against Caspase 3.

The scattered residual CK8+ cell clusters of aneural oral epithelia resembled, by their small size yet expression of terminal differentiation markers, those induced by ectopic expression of *Shh* (*Castillo et al., 2014*). Such clusters might thus represent the endpoint, in the absence of subsequent innervation, of taste bud organogenesis triggered by local Shh (endogenous to the taste placodes, or ectopically induced in the epithelium). Along the same lines (and barring a myriad of possible species differences), the most parsimonious reconciliation of our finding with the apparently opposite one in axolotl would be that the ectopic aneural taste buds obtained in that species are much smaller than normal ones — as appears from Figure 2B versus Figure 4D of *Barlow et al. (1996)*; or that in axolotl, taste buds would be so much smaller than in mouse as to not require a pool of progenitors.

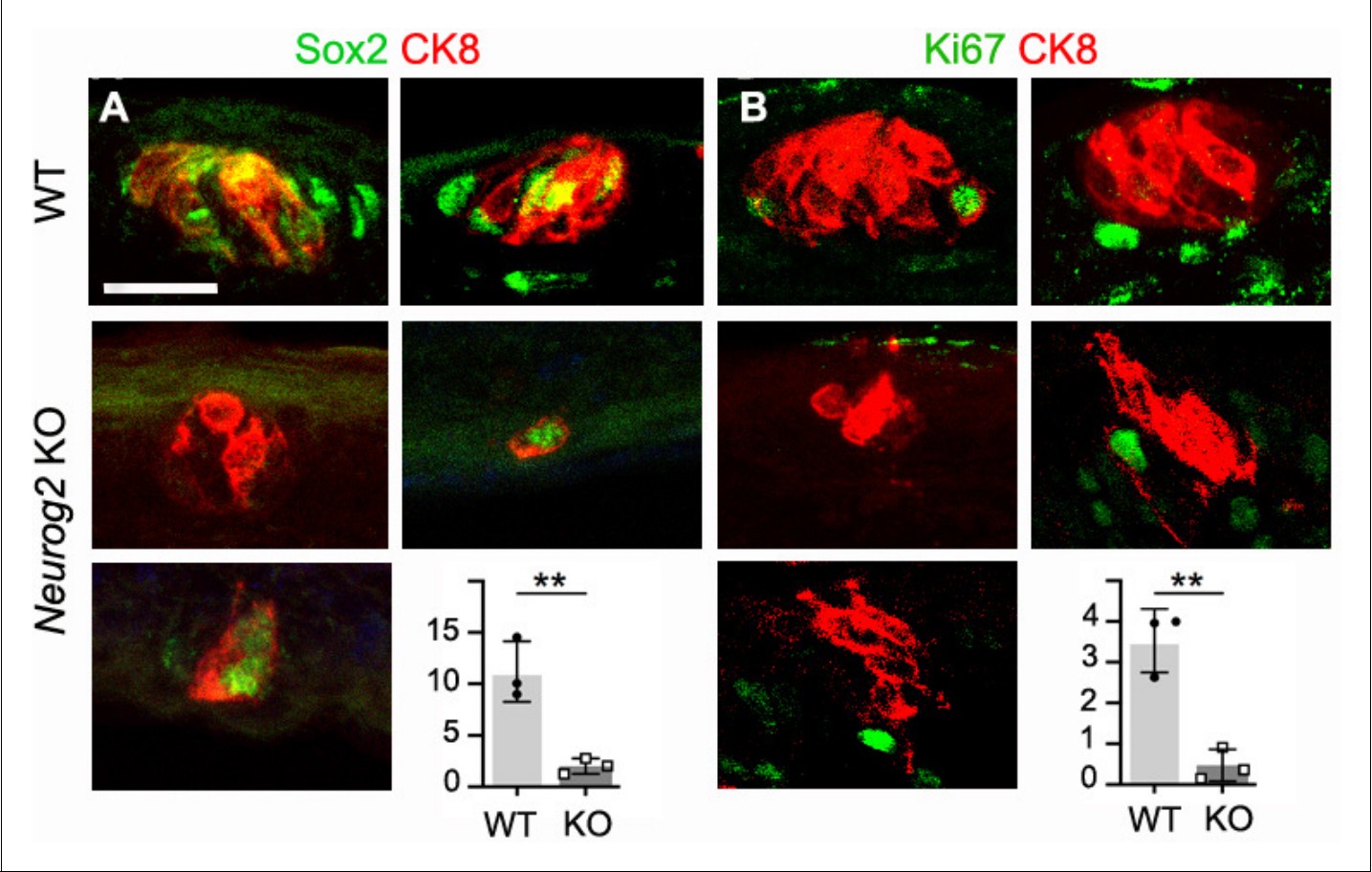

**Figure 4.** Taste nerves are required for Sox2 expression and cell proliferation around taste bud anlagen. (A,B) Immunofluorescence against CK8 and Sox2, or CK8 and the proliferation marker Ki67 on representative CK8$^+$ cell clusters in the soft palate from wild type, and *Neurog2* KO, at E18.5. The graphs show the counts of Sox2$^{High}$ and Ki67$^+$ cells inside or within two cell diameters of CK8$^+$ cell clusters. **: $p<0.01$; error bars presented as mean ± SD (for Sox2$^{High}$ cells, a sample of 72 wild type and all (n = 100) mutant clusters were counted, on three animals; for Ki67$^+$ cells, a sample of 71 wild type and all (n = 101) mutant clusters were counted on three animals). The individual results for each animal are represented by dots. Scale bar: 20 μm.

DOI: https://doi.org/10.7554/eLife.49226.011

The following source data is available for figure 4:

**Source data 1.** Number of Sox2$^+$ and number of Ki67$^+$ cells for each K8$^+$ cell cluster at E18.5 in wild type and *Neurog2* KO pups.
DOI: https://doi.org/10.7554/eLife.49226.012

All in all, we show that formation of taste buds in the mammalian embryo is as dependent on innervation as their renewal and regeneration are in the adult. In line with this, the time between the arrival of nerves and taste bud formation (2 to 3 days in the mouse palate; *Rashwan et al., 2016*) is in the same range as that between re-innervation and taste bud regeneration in gerbils (*Cheal and Oakley, 1977*) — the only species, to our knowledge, where the lag between regrowth of nerves and reformation of taste buds was measured. In light of our results, 'maintenance', in the case of taste buds, can be interpreted as an ongoing ontogeny. The developmental role that we document for nerves is neuron-type specific: exerted by visceral sensory neurons, derived from epibranchial placodes, not by somatosensory neurons, derived from the neural crest. This specificity makes less likely the possibility, that we cannot exclude however, that the nerve associated cells (Schwann cell precursors and Schwann cells) would have the inducing role, rather than the nerve fibers themselves. At the molecular level, nerve-derived *Shh* has been implicated in the maintenance of adult taste buds (*Castillo-Azofeifa et al., 2017*; *Lu et al., 2018*), with a proposed action, however, too slow and too limited to underlie the dramatic effect of nerve cutting on the integrity of taste buds —

or on expression of *Shh* in taste bud cells (*Miura et al., 2004*). The nerve-derived trophic 'hormone-like' substance(s) postulated a century ago (*Olmsted, 1920*) to induce taste buds — in the adult for their continuous renewal, and, as we show here, in the embryo for their ontogeny — are likely yet to be discovered.

# Materials and methods

**Key resources table**

| Reagent type (species) or resource | Designation | Source or reference | Identifiers | Additional information |
|---|---|---|---|---|
| Antibody | Anti- Neurofilament (mouse monoclonal) | Hybridoma Bank | Hybridoma Bank (#2H3) RRID:AB_531793 | one in 500 |
| Antibody | Anti-Gustducin (Gnat3) (goat polycolonal) | MyBioSource | MyBioSource (#MBS421805) RRID:AB_10889192 | one in 500 |
| Antibody | Anti-Entpd2 (rabbit polyclonal) | http://ectonucleotidases-ab.com/ (*Bartel et al., 2006*) | RRID:AB_2314986 | one in 500 |
| Antibody | Anti-Pkd2l1 (rabbit polycolonal) | Millipore | Millipore (#AB9084) RRID:AB_571091 | one in 1000 |
| Antibody | Anti-Prox1 (rabbit polyclonal) | Millipore | Millipore (#AB5475) RRID:AB_177485 | one in 1000 |
| Antibody | Cytokeratin8 (Troma-I) (rat monoclonal) | Developmental Studies Hybridoma Bank (DSHB) | RRID:AB_531826 | one in 400 |
| Antibody | Anti-Kcnq1 (rabbit polyclonal) | Millipore | Millipore (#AB5932) RRID:AB_92147 | one in 1000 |
| Antibody | Anti-Ki67 (rabbite polycolonal) | abcam | abcam (#ab15580) RRID:AB_443209 | one in 200 |
| Antibody | Anti-Sox2 (goat polyclonal) | R and D Systems | R and D Systems (#AF2018) RRID:AB_355110 | one in 500 |
| Antibody | Anti-Trpm5 (guinea pig polyclonal) | Obtained from ER Liman's lab, USC | | one in 500 |
| Antibody | Anti-βIII Tubulin (Tuj1) (mouse monoclonal) | Covance | Covance (#MMS-435P) RRID:AB_2313773 | one in 500 |
| Antibody | Anti-Phox2b (Rabbit polyclonal) | Pattyn et al., Development, 124, 4065–4075 (1997) | | one in 500 |
| Antibody | Donkey anti-goat A488 | Thermo Fisher | Thermo Fisher (#A-11055) RRID:AB_2534102 | one in 500 |
| Antibody | Donkey anti-guinea pig A488 | Jackson Immunoresearch Laboratories | Jackson Immunoresearch Laboratories (#706-545-148) | one in 500 |
| Antibody | Donkey anti-mouse A488 | Invitrogen | Invitrogen (#A-21202) RRID:AB_141607 | one in 500 |
| Antibody | Donkey anti-rabbit A488 | Jackson Immunoresearch Laboratories | Jackson Immunoresearch Laboratories (#711-545-152) | one in 500 |
| Antibody | Donkey anti-rat Cy3 | Jackson Immunoresearch Laboratories | Jackson Immunoresearch Laboratories (#712-165-153) | one in 500 |
| Commercial kit | Vectastain Elite ABC Kits | Vector Laboratories | Vector Laboratories, PK-6101 and PK-6012 | |

*Continued on next page*

*Continued*

| Reagent type (species) or resource | Designation | Source or reference | Identifiers | Additional information |
|---|---|---|---|---|
| Chemical compound | DAB (3,3'-Diaminobenzidine) | Sigma | Sigma (#SLBP9645V) | |
| Commercial reagent | Proligesterone (Delvesterone) | MSD Animal Health | | Delvosteron: NaCl (0.9%)=1:1 |

## Histology

In situ hybridization and immunochemistry on sections or wholemounts have been described (*Coppola et al., 2010*). Immunofluorescence on cryostat or vibratome sections was performed as previously described (*Espinosa-Medina et al., 2014*). Wholemount immunofluorescent staining was made of cleared embryos following the 3DISCO protocol (*Ertürk et al., 2012*) and imaged with an Sp8 confocal microscope. 3D reconstructions were performed using the IMARIS imaging software.

Mouse embryos (E11.5, E12.5, E14.5 and E16.5) or 4% paraformaldehyde (PFA)-perfused pups (at P0 (E18.5) and P2 (E20.5)) were fixed in 4% PFA overnight at 4°C, and the tongue or palate was dissected out. For cryostat sections, tissues were embedded in 7% gelatine/15% sucrose and sectioned at 20 μm. For vibratome sections, tissues were embedded in 3% agarose (in PBS) and cut at 100 μm.

Immunohistochemical reactions were done with the Vectastain Elite ABC Kits (PK-6101 and PK-6012; Vector Laboratories) and color revealed by DAB (3,3'-Diaminobenzidine).

For anti-Pkd2l1 and anti-Sox2 immunoreactions, light fixation was required with 4% PFA at 4°C for 3 hr.

Antisense RNA Probes used were *Hes6* (obtained from Ryoichiro Kageyama), *Ascl1* (obtained from François Guillemot), *Shh* (obtained from Andrew McMahon) and *Sox2* (obtained from Dr. Robin Lovell-Badge).

## Transgenic mouse lines

*Foxg1Cre (BF-1)* KO (*Hébert and McConnell, 2000*): a knock-in of *Cre* in the *Foxg1* locus. RRID: MGI:5806112

*Neurog1* KO (*Ma et al., 1998*): a fragment of the *Neurog1* ORF was replaced by *GFP* and a PGK-neo cassette. RRID:MGI:3639866

*Neurog2* KO (*Florio et al., 2012*): knock in of CreERT2 into the *Neurog2* locus, allowing the expression of a tamoxifen inducible Cre from the *Neurog2* promoter and creating a null allele. RRID:MGI:5432590

*Neurog1/2* KO were bred by intercrossing single *Neurog1* and *Neurog2* heterozygous knockouts, then *Neurog1/2* double heterozygotes.

All lines were maintained by crossing with C57BL/6 x DBA/2 F1 mice.

## Animal treatment

Proligestone (Delvosteron) was used to prolong the gestation of *Neurog1* and *Neurog2* KO mice. At E16.5, 100 μL of diluted Delvosteron (MSD Animal Health) solution (Delvosteron: NaCl (0.9%)=1:1) was injected subcutaneously in each pregnant mouse. Pups were surgically delivered on day 20.5 of embryonic development.

## Statistical analyses

For the quantification of CK8$^+$ cell clusters, one cryosection out of two through the entire soft palate was analyzed in *Neurog1* and *Neurog2* KO embryos, wild types serving as controls. This raw number was used as an estimate of the number of CK8$^+$ cell clusters in the palate, without multiplying by two and performing the Abercrombie correction. Indeed, all the clusters we counted in the wild type spanned the whole height of the epithelium and/or had a visible pore, de facto excluding objects that would be tangentially sectioned. This amounts to a systematic loss of 'caps', an important source of errors in the Abercrombie correction (*Hedreen, 1998*). Our method makes it next to impossible that clusters (with an average width of 28 μm (E18.5) and 38 μm (E20.5), see *Source data 1*) would be counted more than once on alternate 20 μm sections. On the other hand, a few clusters

were probably lost, if they were centered on one of the uncounted section. The accuracy of our method is verified by the fact that we found the same number of clusters at E20.5 and E18.5 despite the 36% increase in average diameter, and also that we found (not shown) the same number of clusters on alternate 30 μm sections (as opposed to 20 μm).

For each genotype, all CK8$^+$ clusters (if there were less than 10), or up to ten (if there were more) were imaged, for the wild type in 2 to 3 sections around the midline of the soft palate (where taste buds were the densest), for the mutants, in one out of two sections throughout the palate. Confocal imaging was performed on a Leica SP5 microscope or a LSM 880 Airyscan Zeiss microscope. The surface occupied by each CK8$^+$ cluster was automatically outlined and measured with the Fiji software, and the mean cross-sectional area was calculated for each genotype. To calculate the entire surface occupied by CK8$^+$ cell clusters on alternate sections of one half of soft palate, the mean value of the calculated surfaces of CK8$^+$ cell clusters was multiplied by the raw, uncorrected number of CK8+ clusters. Statistical analysis was performed using unpaired two-tailed $t$-test. Results are expressed as mean ± SD. All graphs were performed with GraphPad Prism software. For the reason that we did not count 'caps' of wild type CK8+ clusters (see above), the decrease in total area is likely underestimated.

# Acknowledgements

We thank the Imaging Facility of IBENS, which is supported by grants from Fédération pour la Recherche sur le Cerveau, Région Ile-de-France DIM NeRF 2009 and 2011 and France-BioImaging. We wish to thank the animal facility of IBENS, C Goridis for helpful comments on the manuscript and all the members of the Brunet laboratory for discussions. This study was supported by the CNRS, the École Normale Supérieure, INSERM, ANR award 17-CE160006-01 (to J-FB), FRM award DEQ2000326472 (to J-FB), the 'Investissements d'Avenir' program of the French Government implemented by the ANR (referenced ANR-10-LABX-54 MEMO LIFE and ANR-11-IDEX-0001–02 PSL Research University). D-F received a fellowship from the China Scholarship Council. GGC's lab was funded by the Italian Telethon Foundation grant GGP13146.

# Additional information

## Funding

| Funder | Grant reference number | Author |
|---|---|---|
| Agence Nationale de la Recherche | ANR-12-BSV4-0007-01 | Jean-François Brunet |
| Agence Nationale de la Recherche | ANR-10-LABX-54 MEMOLIFE | Jean-François Brunet |
| Agence Nationale de la Recherche | ANR-11-IDEX-0001-02 PSL research University | Jean-François Brunet |
| Fondation pour la Recherche Médicale | DEQ 2000326472 | Jean-François Brunet |
| Agence Nationale de la Recherche | ANR-17-CE16-00006-01 | Jean-François Brunet |
| China Scholarship Council | | Di Fan |

The funders had no role in study design, data collection and interpretation, or the decision to submit the work for publication.

## Author contributions

Di Fan, Conceptualization, Investigation, Writing—original draft; Zoubida Chettouh, Validation, Investigation, Visualization, Methodology, Writing—review and editing, Involved in every step of the production, acquisition and analysis of the data; G Giacomo Consalez, Resources, Methodology, Writing—review and editing, Provided the main genetic tool for this study, the Neurog2 KO; Jean-

François Brunet, Conceptualization, Supervision, Funding acquisition, Visualization, Writing—original draft, Project administration

### Author ORCIDs

Jean-François Brunet (iD) https://orcid.org/0000-0002-1985-6103

### Ethics

Animal experimentation: All animal studies were done in accordance with the guidelines issued by the French Ministry of Agriculture and have been approved by the Direction Départementale des Services Vétérinaires de Paris.

### Decision letter and Author response

Decision letter https://doi.org/10.7554/eLife.49226.016
Author response https://doi.org/10.7554/eLife.49226.017

## Additional files

### Supplementary files

• Source data 1. Average width of K8$^+$ cell clusters in wild type and *Neurog2*KO at E18.5 and E20.5.
DOI: https://doi.org/10.7554/eLife.49226.013

• Transparent reporting form
DOI: https://doi.org/10.7554/eLife.49226.014

### Data availability

All data generated or analysed during this study are included in the manuscript, and in the source data files.

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
