## [Decision Letter]

Thank you for submitting your article "Taste bud formation depends on taste nerves" for consideration by *eLife*. Your article has been reviewed by Marianne Bronner as the Senior Editor, a Reviewing Editor, and three reviewers. The following individual involved in review of your submission has agreed to reveal their identity: Igor Adameyko (Reviewer #3).

The reviewers have discussed the reviews with one another and the Reviewing Editor has drafted this decision to help you prepare a revised submission.

As you will see, all of the reviewers were impressed with the significance and novelty of your work, but each reviewer also had specific and useful comments for improving the manuscript. Specifically, there are suggestions relate to the quantification of the data (reviewers 1 and 2), the over-simplification of the Title and Abstract (reviewers 2 and 3), and the Discussion section (all three reviewers).

I am including the three reviews at the end of this letter. I appreciate that the reviewers' comments cover a range of suggestions for improving the manuscript. We trust that most of these can be accommodated in a reasonable period of time. We look forward to receiving your revised manuscript.

*Reviewer #1:*

This is a nifty short report on genetic ablation of either somatosensory or visceral sensory neurons and the effects on development of taste buds. The work is generally clear and well-written and offers a modicum of new information on the neural dependency of the sensory end organs for taste.

A major obstacle to clarity in this work is the lack of a clear definition of how the authors use the term "Taste buds". As the MS describes, development in this system proceeds from non-innervated placodes through some intermediate stages culminating in a mature taste bud. The terminology used in this work seems inconsistent, sometimes referring to the intermediate stages as "taste anlage", but never clearly defining the term. Does a taste anlagen contain elongate, differentiated taste cells? When does a small collection of elongate taste cells become a "taste bud"? For example, Figure 2 shows elongate differentiated taste cells in Neurog2 KO animals, but the caption suggests that taste buds do not form in this KO line. Since the bar graphs of this figure report on number of taste buds observed, it is essential that the authors define what structure was counted as a taste bud. If in fact the authors are counting as taste buds these small collections of differentiated elongate cells, then they should be correcting for the sampling bias comparing larger objects to smaller objects in histological sections, e.g. by Abercrombie correction. If not, this is essential. Crucial for this calculation is the section thickness and the spacing between sections in relation to the size of the objects being studied. For example, if the section thickness is 15um and the object is 50um in depth, then each object is guaranteed to be counted twice if alternate sections are measured. Conversely, in the same set of sections a 10um object will be counted only once giving the impression that there are twice as many large objects as small objects when in fact, there would be equal numbers.

Also missing from this report is any mention of the development of taste bud primordia (placodes?) in vitro - a situation entirely devoid of innervation. This would seem highly relevant to this paper. Relevant references:

Mbiene, Maccallum and Mistretta, 1997.

Mistretta et al., 2003.

Hall, Bell and Finger, 2003.

Ozdener et al., 2006.

*Reviewer #2:*

Understanding the role of innervation in taste bud development is complicated by the normal occurrence of taste buds in the palate and different parts of the tongue, with taste cells maturing at different rates in each of these locations. Fan et al. exploited their genetic models to dissect the involvement of nerves in taste bud formation at different stages and in distinct locations.

Developmental defects in these mouse mutants include the absence of neurons that normally innervate the taste buds, and these unique neuronal knockouts therefore can be used to assess whether nerves are required in taste bud development. The authors showed that at early stages when nerves are just about to establish contacts with their peripheral targets, the circumvallate placode at the posterior of the tongue is unique in its nerve dependence for expression of certain early genes and for taste bud formation. Expression of some of the same genes during development of non-taste palatal placodes or lingual placodes giving rise to taste buds of the fungiform papillae in contrast remains largely unaffected. Nerves thus participate in taste bud formation at specific developmental windows that vary by location. This variation in nerve dependence for formation of distinct taste bud populations raises questions as to the choice of title, which understates the complexity of taste bud dependence on innervation.

It would be helpful if the authors could provide an overview describing taste bud development and maturation at each location and at distinct developmental stages. Using the Neurog2 knockout alone in palatal taste bud development, a previously unappreciated early involvement of nerves in taste bud maintenance was noted. The authors showed morphological defects in the number of K8+ cells and reduced proliferation and reduced expression of *Sox2* associated with progenitor cells. However, sample numbers and quantification of these observations were not provided in the current manuscript. A revision of the manuscript is recommended before acceptance for publication. Please see below for specific questions and suggestions.

1) The Abstract and Title do not include any reference to the complexity of nerve involvement in formation of taste buds at different locations and developmental stages.

2) Does circumvallate atrophy occur in the Neurog2 single knockout, as was noted for the Neurog1/2 double KO?

3) Please report the number of mutant animals examined in Figure 1.

4) One of the most interesting aspects of the paper is mutant effects on progenitor cells, however, more samples are needed to quantify *Sox2Sox2*^+^ progenitor cells and proliferative status as indicated by Ki67+. Additional time points at E16.5 and E17.5 would also be informative.

*Reviewer #3:*

This focused and strong study answers an important biological question addressing the importance of innervation in initial development of taste buds and putting the regeneration of taste buds into an ontogeny-like framework. The authors provide a clear and straightforward answer in the form of a short report.

I have very few comments:

1) I think that the abstract slightly over-exaggerates the real conclusions: not all taste buds depend on the nerve presence in their development.

2) It should be very briefly discussed why different taste buds demonstrate differential preferences for the specific visceral sensory nerves during their development. Why the embryonic environment is permissive for some buds that are not innervated at all in case of a double Neurog1/2 KO? Can it be connected to the extraordinary high levels of Shh in developing rugi of the palate? This would fit the results described in the paper from Linda Barlow lab: https://www.ncbi.nlm.nih.gov/pmc/articles/PMC4197660/

3) In Figure 2—figure supplement 1: In case of Neurog2 KO, Tuj1+ green fibers are very proximal to the CK8 staining and the location of forming taste bud. Are those nerves the remaining touch afferents? They stay in place in Neurog2 KO, although they do not support the development of fungiform papillae. It will be good if the authors clarify this in a figure legend for pedagogical purposes.

4) Can it be that nerve-associated cells (SCPs and Schwann cells) play a key role in signaling to the developing taste buds, and not the neurons, which support them (although I think this is unlikely)? Worth discussing in one or two phrases.

Overall, this is a good report that requires only some tuning of the text.

---

## [Author Response]

Reviewer #1:

This is a nifty short report on genetic ablation of either somatosensory or visceral sensory neurons and the effects on development of taste buds. The work is generally clear and well-written and offers a modicum of new information on the neural dependency of the sensory end organs for taste.A major obstacle to clarity in this work is the lack of a clear definition of how the authors use the term "Taste buds". As the MS describes, development in this system proceeds from non-innervated placodes through some intermediate stages culminating in a mature taste bud. The terminology used in this work seems inconsistent, sometimes referring to the intermediate stages as "taste anlage", but never clearly defining the term. Does a taste anlagen contain elongate, differentiated taste cells? When does a small collection of elongate taste cells become a "taste bud"?

The referee rightly points to the lack of clear-cut transition during taste bud formation. Indeed, we don’t know of any morphological criterion or global gene expression event that sharply demarcates the three canonical stages of taste bud formation: “placode”, “papilla” (whose mesenchymal core forms progressively), and “taste organ” (i.e. a papilla equipped with a mature taste bud). The expression of *Shh* or *CK8* spans 3 and 2 of these phases, respectively, and their restriction to future taste bud cells is progressive. As for terminal differentiation markers (*TRPM5, Pk2dL1* etc.), the onset of their expression is not precisely known. So, to the question “When does a small collection of elongate taste cells become a "taste bud"?” there is, in our opinion, no rigorous answer.

This said, to homogenize the nomenclature as requested, we now call any group of elongated CK8^+^ cells at E18.5 or later, “CK8^+^ cells clusters” whether they are mature taste buds, immature taste buds (that we formerly called “taste bud anlagen” — there was never mention of “taste anlagen” —, i.e. not yet functional or not containing the full complement of cells), or taste buds which are atrophic in the mutants, to any extent. This common term reflects the fact that all such formations are not distinguished by dichotomous morphological or gene expression criteria, but distributed on a gradient of sizes (and we quantify the sizes).

This leaves undefined the nature of CK8^+^ cells between E16.5 and 18.5, before they are clearly elongated or grouped in onion shaped structures, all the more so, since we show that some of these cells (on the dorsal aspect of the circumvallate papilla and on the rugae) never become taste cells.

Nevertheless, we have kept the word “taste bud” in the title of the paper and figures and in the Discussion section, because a large majority of CK8+ clusters have vanished in the mutants and the remaining ones are abnormally small, so that, despite the lack of definition of the exact moment where a collection of CK8+ cells qualifies as a “taste bud”, we feel justified in saying that “taste bud formation requires taste nerves” [see responses to criticisms of the title below].

For example, Figure 2 shows elongate differentiated taste cells in Neurog2 KO animals, but the caption suggests that taste buds do not form in this KO line. Since the bar graphs of this figure report on number of taste buds observed, it is essential that the authors define what structure was counted as a taste bud.

See explanation above. We have now replaced “taste buds” by “CK8^+^ cell clusters” in the legend and in the figure. And we counted all CK8^+^ cell clusters (down to single cells when that occurred in the mutants).

If in fact the authors are counting as taste buds these small collections of differentiated elongate cells, then they should be correcting for the sampling bias comparing larger objects to smaller objects in histological sections, e.g. by Abercrombie correction. If not, this is essential. Crucial for this calculation is the section thickness and the spacing between sections in relation to the size of the objects being studied. For example, if the section thickness is 15um and the object is 50um in depth, then each object is guaranteed to be counted twice if alternate sections are measured. Conversely, in the same set of sections a 10um object will be counted only once giving the impression that there are twice as many large objects as small objects when in fact, there would be equal numbers.

We do not think that the Abercrombie correction, which was devised to count cell nuclei in a volume of tissue, is appropriate for the comparison of very different objects between a wild type and mutant condition, particularly because of the indirect estimate of H (the height of the object perpendicular to the plane of section) and the “lost caps” effect We now make this reasoning explicit in the Material and methods section: in the following way:

“This raw number was used as an estimate of the number of CK8^+^ cell clusters in the palate, without multiplying by two and performing the Abercrombie correction. Indeed, all the clusters we counted in the wild type spanned the whole height of the epithelium and/or had a visible pore, de facto excluding objects that would be tangentially sectioned. This amounts to a systematic loss of “caps”, an important source of errors in the Abercrombie correction (Hedreen, 1998). Our method makes it next to impossible that clusters (with an average width of 28μm(E18.5) and 38μm (E20.5)) would be counted more than once on alternate 20μm sections. On the other hand, a few clusters were probably lost, if they were centered on one of the uncounted section. The accuracy of our method is verified by the fact that we found the same number of clusters at E20.5 and E18.5 despite the 36% increase in average diameter, and also that we found (not shown) the same number of clusters on alternate 30μm sections (as opposed to 20μm).”

In any case, this raw number is the only one relevant to the calculation of the total volume occupied by CK8+ cells, that we estimate through the proxy of the total surface occupied by CK8+ cells on the sections we count (one 20μm section out of two). The total surface occupied by taste cells is the total number of patches of CK8+ cells seen on sections, multiplied by the average size of the patches. If two patches on two sections happened to correspond to the same “CK8+ cell cluster”, it would be a realistic reflection of the fact that this cluster is big and occupies a large volume (which is what we evaluate, by a proxy). This is biologically the most important figure. If the number of CK8+ clusters was unchanged in the mutants, but their size decreased on average by 98%, the conclusion of our paper, i.e. that “taste bud formation depends on taste nerves” would remain unchanged.

*Also missing from this report is any mention of the development of taste bud primordia (placodes?)* in vitro *- a situation entirely devoid of innervation. This would seem highly relevant to this paper. Relevant references:*

Mbiene, Maccallum and Mistretta, 1997.Mistretta et al., 2003.Hall, Bell and Finger, 2003.Ozdener et al., 2006.

We did cite the first paper, historically, to show that fungiform papillae develop in tongue explants. As requested, we have now added the following citation:, Maccallum and Mistretta, 1997 and Hall, Bell and Finger, 2003. And we added Mistretta et al., 2003 on the topic of the CV papilla, making explicit that we find, like her, that Shh is switched on in the CV placode, but leaving implicit the contradiction of her claim that CV papilla formation is nerve independent (on the basis of morphological evidence that we think is weak). On the other hand, it is not easy to see how the taste cell culture system described by Ozdener et al., 2006 informs about nerve dependency in vivo.

Reviewer #2:

[…] It would be helpful if the authors could provide an overview describing taste bud development and maturation at each location and at distinct developmental stages.

The circumvallate placode is unique in its dependency on nerves for subsequent papilla formation, but not for taste bud formation, which are just as dependent as in other papillae. To help conceptually disentangle the two issues, we now show, the absence of taste buds in the neonatal circumvallate organ (as expected since the organ itself does not form) (Figure 2—figure supplement 1).

Using the Neurog2 knockout alone in palatal taste bud development, a previously unappreciated early involvement of nerves in taste bud maintenance was noted. The authors showed morphological defects in the number of K8+ cells and reduced proliferation and reduced expression of Sox2 associated with progenitor cells. However, sample numbers and quantification of these observations were not provided in the current manuscript. A revision of the manuscript is recommended before acceptance for publication.

We do not document “maintenance” but appearance.

Please see below for specific questions and suggestions.1) The Abstract and Title do not include any reference to the complexity of nerve involvement in formation of taste buds at different locations and developmental stages.

The “complexity” (which is rather an asynchrony) is in the development itself, not in the requirement for nerves in that development. The development of taste buds is not synchronous at every place (CV, soft palate, and tongue, and even different parts of the anterior tongue) and their demise at different stages in the absence of nerves simply reflects that.

The only true “complexity”, not encapsulated by the Title or the Abstract, is that the CV organ depends on nerves, not only for its resident taste buds but also for the papilla itself. In other words, it depends on the nerves even more than other taste organs. The title does not include this additional feature, but remains perfectly true, concerning taste buds.

We think that one virtue of our title (and of the paper), is precisely to simplify a field — which has become replete, over decades, with appearances of complexities and special cases, quantification of partial phenotypes, at different phases, in different locations, etc. — by encapsulating the fact that no complete, mature taste bud forms in the absence of nerves. This most important message would be jeopardized by the addition of what is basically a bonus, the idiosyncrasy of the circumvallate papilla (again, not of its taste buds).

2) Does circumvallate atrophy occur in the Neurog2 single knockout, as was noted for the Neurog1/2 double KO?

As requested, we now show in a new supplementary figure that the *Neurog2* KO looks exactly like the double *Neurog2/Neurog1* knock out. This is now stated in the text as: “A similar phenotype was obtained in single *Neurog2* KO (Figure 1—figure supplement 3A,B), which lack a petrosal ganglion (Figure 1—figure supplement 3C).

3) Please report the number of mutant animals examined in Figure 1.

Two animals were used for each probe. This is now stated in the legend.

4) One of the most interesting aspects of the paper is mutant effects on progenitor cells, however, more samples are needed to quantify Sox2Sox2^+^ progenitor cells and proliferative status as indicated by Ki67+. Additional time points at E16.5 and E17.5 would also be informative.

As requested, we now counted *Sox2*^+^ cells and Ki67 cells in mutants and wild type ad E18.5, and add this data to Figure 4. On a pilot experiment (that we do not show), we could not see any difference at E16.5.

Reviewer #3:

This focused and strong study answers an important biological question addressing the importance of innervation in initial development of taste buds and putting the regeneration of taste buds into an ontogeny-like framework. The authors provide a clear and straightforward answer in the form of a short report.I have very few comments:1) I think that the abstract slightly over-exaggerates the real conclusions: not all taste buds depend on the nerve presence in their development.

It is not clear to us what are the taste buds that the referee has in mind that would not depend on nerves. Unless the referee means that our statement would be justified only if the disappearance of taste cells was by 100% as opposed to 96%? Also, we have no indication that the atrophic residual CK8^+^ cell clusters in the mutants are the same, from animal to animal. We believe that our short and clear abstract reflects the findings that we describe, without exaggeration, and that the field will benefit from that clear message [see response to referee #2]. Nevertheless, we softened the abstract as requested, it now reads “we show that the latter but not the former are absolutely required for the proper formation of their target organs, the taste buds”, leaving room for the formation of very rudimentary, atrophic organs.

2) It should be very briefly discussed why different taste buds demonstrate differential preferences for the specific visceral sensory nerves during their development.

Taste buds per se don’t demonstrate differential preferences (see response to referee#2). Only the circumvallate *papilla* shows a specific dependency that other papillae don’t.

Why the embryonic environment is permissive for some buds that are not innervated at all in case of a double Neurog1/2 KO? Can it be connected to the extraordinary high levels of Shh in developing rugi of the palate? This would fit the results described in the paper from Linda Barlow lab: https://www.ncbi.nlm.nih.gov/pmc/articles/PMC4197660/

We do not see true “taste buds” (i.e. CK8+ clusters which would not be atrophic) in the mutants. The relationship of the atrophic residual taste buds (that we call now CK8+ cell clusters) to the ectopic ones in the Barlow paper is made explicit in the text. (And the rugae never contain taste buds: their CK8+ cells are not taste cell precursors, or possibly are taste cell precursors that never differentiate).

3) In Figure 2—figure supplement 1: In case of Neurog2 KO, Tuj1+ green fibers are very proximal to the CK8 staining and the location of forming taste bud. Are those nerves the remaining touch afferents? They stay in place in Neurog2 KO, although they do not support the development of fungiform papillae. It will be good if the authors clarify this in a figure legend for pedagogical purposes.

In the original main text we wrote: “In each single *Neurog1* and *Neurog2* knockout at E16.5, the epithelium of the tongue and of the soft palate retained nerve fascicles at regularly spaced locations (corresponding to the taste organs) from, presumably, visceral or somatic sensory fibers, respectively (which thus navigate to their target independently of each other)”. As requested, we have now added the following sentence in the legend of Figure 1—figure supplement 1: “Since *Neurog2* KO have lost the geniculate, but keep the trigeminal ganglion, the residual innervation of taste organs in these mutants correspond to the somatic (touch and pain) fibers”.

4) Can it be that nerve-associated cells (SCPs and Schwann cells) play a key role in signaling to the developing taste buds, and not the neurons, which support them (although I think this is unlikely)? Worth discussing in one or two phrases.

As requested we now mention this possibility in the text as follows: “This specificity [of nerve fiber type] makes less likely the possibility, that we cannot exclude however, that the nerve associated cells (Schwann cell precursors and Schwann cells) have the inducing role, rather than the nerve fibers themselves.”